# Absolute excited state molecular geometries revealed by resonance Raman signals

Giovanni Batignani [1,2,5] ✉, Emanuele Mai [1,2,5], Giuseppe Fumero [1], Shaul Mukamel [3] & Tullio Scopigno[1,2,4] ✉

Ultrafast reactions activated by light absorption are governed by multi-dimensional excited-state (ES) potential energy surfaces (PESs), which describe how the molecular potential varies with the nuclear coordinates. ES PESs ad-hoc displaced with respect to the ground state can drive subtle structural rearrangements, accompanying molecular biological activity and regulating physical/chemical properties. Such displacements are encoded in the Franck-Condon overlap integrals, which in turn determine the resonant Raman response. Conventional spectroscopic approaches only access their absolute value, and hence cannot determine the sense of ES displacements. Here, we introduce a two-color broadband impulsive Raman experimental scheme, to directly measure complex Raman excitation profiles along desired normal modes. The key to achieve this task is in the signal linear dependence on the Frank-Condon overlaps, brought about by non-degenerate resonant probe and off-resonant pump pulses, which ultimately enables time-domain sensitivity to the phase of the stimulated vibrational coherences. Our results provide the tool to determine the magnitude and the sensed direction of ES displacements, unambiguously relating them to the ground state eigenvectors reference frame.

Photochemical and photophysical reactions are ruled by excited state (ES) potential energy surfaces (PESs), which drive nuclear motions and initiate the molecular rearrangements along specific vibrational degrees of freedom. The nuclei respond to photo-excitation altering their equilibrium positions, i.e., the molecule undergoes a geometrical rearrangement which, in turn, modifies the physical and chemical properties of the system. Nature has tailored electronically excited PESs in which the vibrational structures are specifically modified from the ground state (GS) equilibrium configuration to efficiently convert the absorbed light energy into molecular rearrangements, driving the biological function and the photo-chemistry by bond length modifications, torsional re-orientations, formation or rupture of chemical bonds. These can be rationalized by the multidimensional displacement $D$ between PESs, i.e., the vector identifying the new equilibrium position in the ES projected onto the eigenvectors ($Q_j$) of the GS with respect to its minimum. Large absolute values of specific displacement components ($d_j = D \cdot Q_j$, commonly referred to as *displacements*) identify the vibrational eigenmodes which are more involved in the excitation and eventually their couplings to the specific electronically excited state. Most importantly, the *sensed* direction of $D$ determines whether the changes in the equilibrium bond distances and angles move closer or away two functional groups, ruling the physical-chemical properties on the ES and, thus, its knowledge is of utmost importance.

For example, while the ultrafast olefinic photoisomerization of stilbene is determined by an increase of the central C=C bond length with out-of-plane motion of the two ethylenic hydrogens[1], the mechanism of vision is habilitated by a sub-picosecond retinal isomerization of the rhodopsin molecule, initially driven by positive and negative changes of different dihedral angles[2]. In diaryl thiophenes, a different ES dynamics is underpinned by an increase of the distance between the carbon (connected to phenyl) and the sulfur atoms in the

[1]Dipartimento di Fisica, Sapienza Universitá di Roma, Roma, Italy. [2]Istituto Italiano di Tecnologia, Center for Life Nano Science @Sapienza, Roma, Italy. [3]Department of Chemistry, University of California, Irvine, CA, USA. [4]Istituto Italiano di Tecnologia, Graphene Labs, Genova, Italy. [5]These authors contributed equally: Giovanni Batignani, Emanuele Mai. ✉e-mail: giovanni.batignani@uniroma1.it; tullio.scopigno@uniroma1.it

thiophene as opposed to a decrease of the dihedral angle between phenyl and thiophene rings[3,4]. The displacement sense also plays a role in the excited states of conjugated push-pull systems, where the alternation of expansion and contraction in specific bond lengths is suggestive of an enhanced zwitterionic character[5]. Finally, the design of molecular motors requires an asymmetry in the excited state PES along the dihedral angle with respect to the ground state, which can be provided, for instance, by chirality, in order to achieve a preferential direction and obtain photoinduced molecular rotary motion[6,7]. Measuring excited state displacements is also important to validate quantum chemistry models for nonlinear optical effects in molecular compounds. These examples demonstrate the importance of determining the sensed direction of the displacements within the Franck Condon (FC) description of the excited state. This information, however, is highly convoluted and difficult to access from the static spectra obtained by absorption or other linear techniques. In particular, reorientation along opposite senses often leads to very similar experimental signatures, motivating a quest for suitable methods to access this quantity.

Spontaneous Raman spectroscopy represents a powerful technique to study the vibrational spectrum of molecular and solid state systems in the GS. Tuning the excitation wavelength to match the energy of an allowed electronic transition to a given ES enables resonant Raman (RR) spectroscopy[8–16] and results in a cross section proportional to the square modulus of the product between the electronic transition moments and the vibrational GS-ES overlaps[11,17–22], ultimately related to their *relative* PES geometries. This fundamentally limits the RR sensitivity uniquely to the absolute values[23] of the ES displacements[24,25]. The missing sign information hampers elucidating the molecular reorientations at the basis of photo-physical and photo-chemical phenomena[10]. In addition, measuring and isolating the RR response under identical conditions for a sequence of narrowband experiments is often a challenging task, further hampered by the fluorescence emission, which typically overwhelms the weak spontaneous Raman signatures.

Nonlinear time-domain Raman schemes, in particular impulsive stimulated Raman scattering (ISRS)[1,26–31], offer the chance to circumvent some of the spontaneous Raman limitations. ISRS employs a pair of temporally well separated pulses to investigate vibrational matter properties: two interactions with a femtosecond Raman pump (RP) generate vibrational coherences, triggering nuclear wavepacket motions and modulating the sample transmissivity, which is then measured by a broadband probe pulse (PP) as a function of its temporal delay $\Delta T$ with respect to the RP. This technique records the Raman response directly in the time-domain, thereby simultaneously accessing the amplitude and the phase of the vibrational oscillations[32–43]. Fast Fourier transforming (FFT) the signal with respect to $\Delta T$ recovers the conventional frequency-domain Raman spectrum. In view of the heterodyne detection, the measured signal is generated on top of the PP and hence the incoherent isotropic fluorescent background is efficiently suppressed[20,44]. Moreover, the time-domain acquisition, performed with temporally separated pump and probe pulses, ensures a Raman response free from nonlinear artifacts caused by cross-phase modulation[30,45,46], which typically affects frequency-domain stimulated Raman approaches[46–48].

In this work, we introduce an experimental protocol, based on a two-color broadband impulsive stimulated Raman scattering (ISRS) scheme, that can determine the real and imaginary parts of the nonlinear RR susceptibility, revealing *absolute* ES geometries.

## Results and discussion
### Design of a two-color ISRS scheme to measure the sense of ES displacements
To take advantage of the phase sensitivity of the ISRS response, we analyze the signal in the time-domain before Fourier transforming,

since evaluating the squared absolute value of the FFT spectra loses the phase information contained in the time-domain data. We employ an off-resonant pump to induce vibrational coherences only on the initially populated electronic level and a probe pulse resonant with a targeted electronic transition. Importantly, the off-resonant pump is critical in that it ensures a stimulation of the vibrational coherences via the molecular polarizability, and not via the Franck-Condon overlap integrals, which are probed solely by the PP. Hence, under the adopted scheme, the ISRS signals are proportional, through the off-resonant GS polarizability, to the amplitude of the GS-ES Franck-Condon overlap integrals and not to their square modulus[49], as opposed to spontaneous Raman spectroscopy. This approach makes the RR ISRS response, monitored as a function of the PP wavelength, sensitive to the sign of PES displacements. The concept is demonstrated in Fig. 1 for the illustrative case of a diatomic molecule, where the unique degree of freedom is the distance between the nuclei, which can be either increased or decreased on the ES (panel a). Importantly, ES molecular geometries corresponding to an increase or a decrease of the nuclear bond-length have FC overlaps with the same magnitude and opposite sign. Hence, the ISRS signal, recorded with the two-color non degenerate scheme presented in this work (panel b), is antisymmetric with respect to the GS-to-ES atomic distance modification (panels c-d) and directly reveals the ES displacement. It is worth to stress that, in principle, the assignment of the sign for a given ground state eigenvector $\boldsymbol{Q}$, obtained by diagonalizing the Hessian matrix, is arbitrary in the sense that each arrow identifying the direction of an atomic vibration may be reversed ($\boldsymbol{Q'} = -\boldsymbol{Q}$, panel e). Importantly, since the proposed ISRS scheme accesses the product between the GS polarizability derivative $\frac{\partial \alpha}{\partial Q}$ and the FC factors, reversing the sign for a given ground state eigenvector changes both the sign of the polarizability derivative as well as the sign of the FC overlaps (panels f-g), making the experimental signals invariant on the selection of such reference frame. Hence, at variance with the spontaneous Raman approach, the sign of the ES displacements relative to the GS eigenvectors can be defined and the ES equilibrium configuration along a given vibrational mode is uniquely identified from the measured ISRS Raman excitation profiles (REPs).

The broadband ISRS response can be evaluated -in a semiclassical framework- through a perturbative expansion of the molecular density matrix in powers of the classical electric fields $E(t) = \sum_i \mathcal{E}_i(t)e^{-i\omega_i t} + c.c.$, using a diagrammatic approach to visualize the concurring processes that generate the time-domain Raman signal[44,50,51]. Particularly, since the RP can act both on the ket and on the bra side of the density matrix, two classes (A/B) of Feynman diagrams (reported in Fig. 2a, b) have to be considered to calculate the third-order polarizabilities, $P_A^{(3)}(\omega, \Delta T)$ and $P_B^{(3)}(\omega, \Delta T)$ respectively, that originate the system response[51,52]. For small modifications of the PP across the sample, the spectrally dispersed ISRS signal $S(\omega, \Delta T)$, defined as the normalized difference between the transmitted PP spectrum in presence ($I_P^{RP\,On}(\omega, \Delta T)$) and in absence ($I_P(\omega)$) of the RP, can be expressed as the imaginary part ($\Im$) of the ratio between the total third-order polarization $P^{(3)}(\omega, \Delta T) = P_A^{(3)}(\omega, \Delta T) + P_B^{(3)}(\omega, \Delta T)$ and the probe field spectral envelope $E_P(\omega)$, accordingly to

$$S(\omega, \Delta T) = \frac{I_P^{RP\,On}(\omega, \Delta T) - I_P(\omega)}{I_P(\omega)} \propto -\Im\left[P^{(3)}(\omega, \Delta T)/E_P(\omega)\right] \quad (1)$$

It is worth to stress that, as the RR process stimulated by the probe pulse projects the vibrational wavefunction onto the ES manifold, both the $P_A^{(3)}(\omega, \Delta T)$ and the $P_B^{(3)}(\omega, \Delta T)$ terms contain sum-over-states expressions[44,53], and are therefore sensitive to the ES geometry.

For an electronically off-resonant RP, the radiation–matter interaction Hamiltonian involved in the preparation of the vibrational

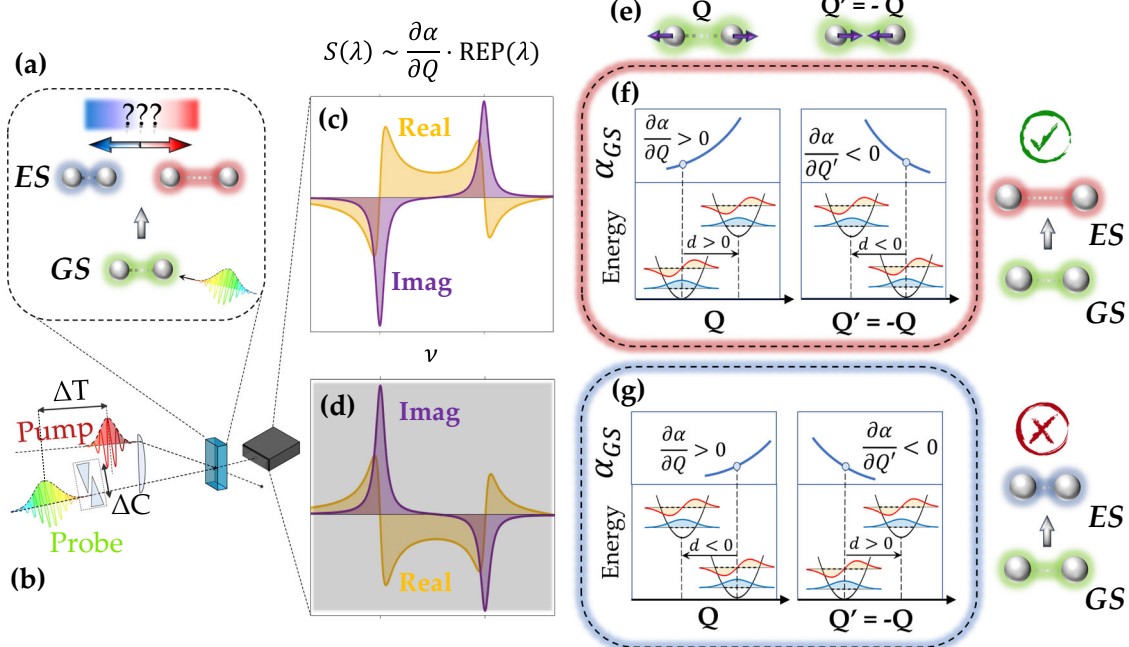

**Fig. 1 | Concept of the proposed Impulsive Stimulated Raman Scattering (ISRS) scheme for a diatomic molecule.** A pair of ultrashort pulses is exploited to measure the complex Raman excitation profile (REP) and access the excited state (ES) displacement, determining whether the atomic distance is increased or decreased upon the photoexcitation **a**, **b**. Since the ISRS response is proportional to the product between the ground state (GS) polarizability derivative and the Franck-Condon (FC) overlaps, the measured signals $S(\lambda)$ uniquely determines the ES geometry and the sense of the geometrical rearrangement **c**, **d**, being antisymmetric with respect to the nuclear distance modification. Notably, while the sense of the GS eigenvector is arbitrary, as both outgoing ($Q$) or in-going arrows ($Q' = -Q$) can be selected **e**, reversing such a sense changes both the sign of the polarizability derivative as well as the sign of the FC overlaps **f**, **g**, making the experimental signals invariant on the selection of the reference frame. For example, if the complex REP reconstructed from the ISRS signal is the one reported in **c**, the consequences of the two possible eigenmode choices **e** are depicted in the corresponding **f**: the eigenmode direction $Q$ is associated to a positive $\frac{\partial \alpha}{\partial Q}$ and a positive displacement $d$, while the direction $Q' = -Q$ corresponds to a negative polarizability derivative and a negative $d$. Crucially, the two formal descriptions yield to the same physical observable, i.e., an increase of the bond length in the excited state (highlighted in red). On the other hand, the measured REP in **c** excludes an ES bond-length shortening (blue), as the corresponding descriptions shown in **g** are both associated to the signal in **d**, which has a reversed sign with respect to the measured one.

coherences is[54] $H_I^{(RP)} = -\alpha \cdot |E_R|^2$, where $\alpha$ indicates the molecular polarizability, while for the resonant PP detection it reads as $H_I^{(PP)} = -\mu \cdot E_P$, where $\mu$ is the dipole operator. Indicating the frequency difference between the $j$-$i$ levels as $\omega_{ji} = \omega_j - \omega_i$, the third order polarizations related to the $P_A^{(3)}(\omega, \Delta T)$ and $P_B^{(3)}(\omega, \Delta T)$ terms can be written as

$$\begin{cases} P_A^{(3)}(\omega, \Delta T) = -\frac{i}{\hbar^3} e^{-i\omega_{g'g}\Delta T} E_P(\omega - \omega_{g'g}) V P_{g'}^{RP} R_{g'}^S(\omega) \\ P_B^{(3)}(\omega, \Delta T) = \frac{i}{\hbar^3} e^{i\omega_{g'g}\Delta T} E_P(\omega + \omega_{g'g}) V P_{g'}^{RP} R_{g'}^{AS}(\omega) \end{cases} \quad (2)$$

where

$$V P_{g'}^{RP} = \frac{\hbar}{2} \frac{\partial \alpha}{\partial Q_{g'}} \langle g'|Q|g\rangle \int \frac{d\omega'}{2\pi} E_R^0(\omega') E_R^{0*}(\omega' - \omega_{g'g}) \quad (3)$$

is the vibrational coherence preparation function (that is the same for the two A/B processes for an off-resonant pump, as detailed in Supplementary Note 1), which depends on the Raman field $E_R^0(\omega)$ and on the molecular polarizability via $\frac{\partial \alpha}{\partial Q_{g'}}$. The absolute value of the latter can be conveniently obtained by an off-resonant Raman experiment, while its sign encodes the structure of the GS eigenvectors which, in turn, define the normal modes reference frame. Importantly, the information regarding the excited state displacement (along the considered $\omega_{g'g}$ normal mode) is fully contained in the $R_{g'}^S(\omega)$ and $R_{g'}^{AS}(\omega)$ REPs, which are probed respectively by the $P_A^{(3)}$ and the $P_B^{(3)}$ pathways. Since the $R_{g'}^{AS}$ term is shifted by one vibrational quantum with respect to $R_{g'}^S$, they corresponds to the anti-Stokes and Stokes

complex Raman excitation profiles:

$$R_{g'}^S(\omega) = \sum_k \frac{\mu_{g'e_k}\mu_{e_kg}}{\omega - \tilde{\omega}_{e_kg}}, \quad R_{g'}^{AS}(\omega) = \sum_k \frac{\mu_{g'e_k}\mu_{e_kg}}{\omega - \tilde{\omega}_{e_kg'}} = R_{g'}^S(\omega - \omega_{g'g}) \quad (4)$$

where the summation over $k$ in Eq. (4) takes into account the ES vibrational manifold and $\tilde{\omega}_{ji} = \omega_{ji} - i\Gamma_{ji}$, which depends on the dephasing rate $\Gamma_{ij} = (T_{vib}^{ji})^{-1}$ of the $|j\rangle\langle i|$ coherence. It is worth to stress that, while the A/B diagrams reported in Fig. 2a, b do not correspond to the spontaneous Raman Stokes/anti-Stokes processes, the square moduli of the REP reported in Eq. (4) represent the information that can be accessed by spontaneous Raman spectroscopy[11]. Crucially, $R_{g'}^S(\omega)$ and $R_{g'}^{AS}(\omega)$ include the product of two dipole matrix elements, namely $\mu_{g'e_k}\mu_{e_kg}$. As shown in Fig. 2d, this product is an odd function of the displacement between the ground and the excited PESs along the considered normal mode: hence, accessing the sign of the real/imaginary part of the REPs, which depend linearly on the $\mu_{g'e_k}\mu_{e_kg}$ product (cf. Fig. 2e), provides the chance for unambiguously determining the sign of ES displacements along the considered normal modes.

Based on the odd symmetry of the complex valued Raman excitation profiles with respect to the displacement (Fig. 2e), the sign of this latter can be in fact identified by the ISRS signal. Aiming at directly accessing the real and imaginary REPs components, which in the measured signal are entangled in a linear combination (see Eq. (6) in the Methods section), an appropriate tuning of the probe chirp ($C_2$, related to the group delay dispersion $GDD$ as $C_2 = GDD/2$) can be directly exploited. Indeed, since the $P_A^{(3)}$ and $P_B^{(3)}$ terms originate from

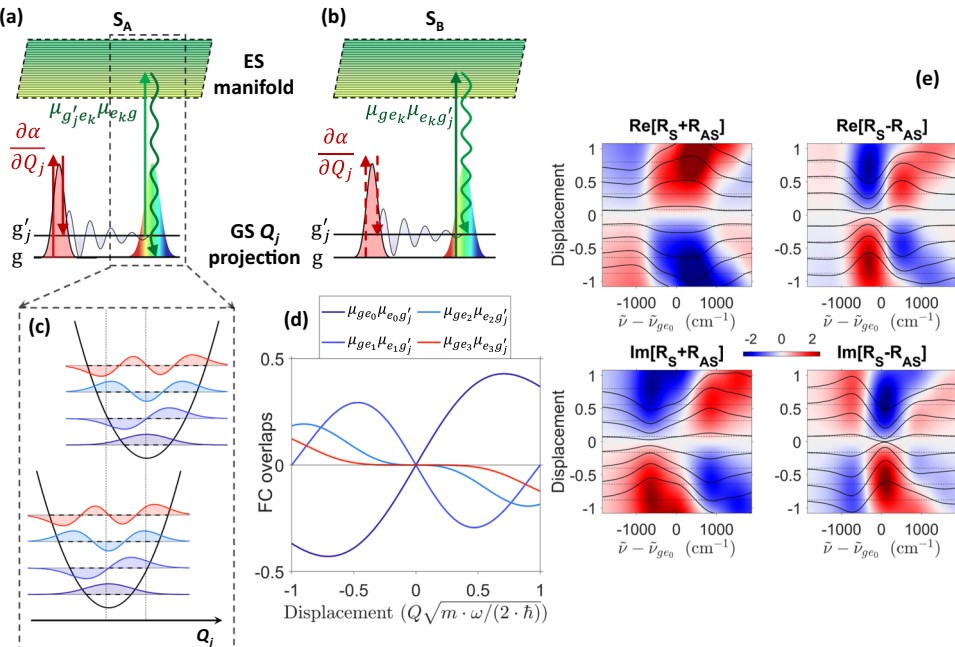

**Fig. 2 | Schematic of the ISRS third-order nonlinear processes.** Energy level diagrams accounting the for impulsive stimulated Raman processes are reported in **a**, **b**. Two consecutive interactions with the RP, which exerts a driving force on the vibrational normal mode $Q_j$ with a coupling ruled by the electronic polarizability derivative $\frac{\partial \alpha}{\partial Q_j}$, prepare a vibrational coherence on the ket **a** or on the bra side **b** of the density matrix. Interactions with the ket/bra side of the density matrix and the (classical) electromagnetic fields are presented by continuous/dashed vertical arrows, respectively. The sample density matrix is then promoted on the electronic excited state (ES) manifold $e_k$ via an interaction, proportional to FC overlaps

$\mu_{ge_k}\mu_{e_kg'}$, with the broadband resonant probe pulse, enabling to record upon a free induction decay the ground state (GS) Raman response in the time-domain. One-dimensional projections of the ES PES along a specific normal mode are reported in **c**. Since the a/b pathways responsible for the measured signal depend on the product between two dipole moments $\mu_{ge_k}\mu_{e_kg'}$ (reported in **d** for a single mode) and not on their absolute squared value, the experimental response, reported in arbitrary units in **e**, is sensitive to the linear combination of complex Stokes/anti-Stokes REPs ($R_S$ and $R_{AS}$, respectively), which reveal the sign of the molecular displacement.

interactions with different PP spectral components[52,55,56], $C_2$ introduces a desired complex phase factor between the A and B pathways, providing the chance to directly measure the displacement sense and the real/imaginary parts of the Stokes/anti-Stokes complex REPs. As summarized in Table 1 and detailed in the Methods section, two values of the probe chirp, namely $C_2 = \frac{m}{2\pi}T_{vib}^2$ and $C_2 = \frac{1}{2\pi}\left(m + \frac{1}{4}\right)T_{vib}^2$ (with $m = 0, \pm 1, \pm 2, \ldots$), and two values of the dechirped temporal delay $\tilde{T} = n\,T_{vib}$ and $\tilde{T} = \left(n + \frac{1}{4}\right)T_{vib}$ (corresponding to $\phi = 2n\pi$ and $\phi = 2\left(n + \frac{1}{4}\right)\pi$, respectively, with $n = 0, 1, 2, \ldots$) are required to obtain the real and the imaginary parts of $R_{g'}^S(\omega)$ and $R_{g'}^{AS}(\omega)$.

## Impulsive Raman experiments and reconstruction of complex REPs

As a road test for the capabilities of the presented approach, we applied the two-color broadband ISRS scheme to study the excited state PES of Rhodamine B (RhB), a staining fluorescent dye extensively used for chemical and biological applications. In view of its high fluorescence quantum yield[57] and small Stokes-shift[58], the fluorescent background overwhelms the spontaneous Raman response, which hence can be efficiently measured only for pump excitation wavelengths far from the absorption maximum[59].

The ISRS spectra of RhB dissolved in methanol have been measured at ambient temperature as a function of the PP wavelength $\lambda_P$, for $\Delta T \in [-0.3, 3]$ ps, with a 10 fs time-step; in Fig. 3a, b we report the time-domain traces recorded using a PP with a vanishing chirp $C_2 \approx 0$. The RP is tuned to be electronically off-resonant ($\approx 700$ nm) and stimulates the normal modes with a not vanishing polarizability derivative, while the PP spectral profile covers the entire sample absorption (cf. Fig. 3e). By fast Fourier transforming over the RP-PP time delay, the frequency-domain spectrum (reported in Fig. 3c, d) is extracted, identifying a strong mode at 620 cm$^{-1}$ and other weaker Raman bands at 490, 735, 1280 and 1360 cm$^{-1}$. Indeed, the ~ 18 fs temporal duration of the pump pulse guarantees efficient stimulation of coherences up to 1400 cm$^{-1}$.

In order to reconstruct the time-domain vibrational response of the single Raman modes, the ISRS traces are further fine scanned around $\Delta T \in [0.35, 0.85]$ ps with a 4 fs time-step, and then fitted as the sum of sinusoidal terms, by the equation $S(\lambda, \Delta T) = \sum_k A_k(\lambda) \sin\left[\omega_{g'_k g}\Delta T + \phi_k(\lambda)\right]$. The frequencies $\omega_{g'_k g}$ are fixed parameters, determined from the peak positions of the FFT spectra, while the extracted amplitudes $A_k(\lambda)$ and phases $\phi_k(\lambda)$ are reported in Fig. 4 for different probe chirps ($C_2$) as a function of $\lambda = \lambda_P$. We note that for $C_2 \approx 0$, all the modes exhibit a $\pi$ phase shift around the absorption maximum at 550 nm, where the oscillation amplitude vanishes.

The fitted $A_k(\lambda)$ and $\phi_k(\lambda)$ parameters were then used to reconstruct the time-domain ISRS response of the different vibrational modes. In Fig. 5a, b, this is evaluated for the 735 cm$^{-1}$ mode, testifying the sensitivity of the scheme also to isolate weak Raman signals (see Fig. 3c, d). Here two PP chirp values are considered, namely $C_2 = -4$ fs$^2$ and $C_2 = 71$ fs$^2 \approx \frac{1}{8\pi}T_{vib}^2$, which closely correspond to the values reported in Table 1 (with $m = 0$) for separately accessing the real and imaginary parts of the mode complex REP. In panels c, d, the signal is then evaluated as a function of the dechirped temporal delay $\tilde{T}$, while four slices of such maps at $\tilde{T}/T_{vib} = n$ and $\tilde{T}/T_{vib} = (n - 1/4)$ are reported in

**Table 1 | By properly tuning the RP-PP temporal delay $\Delta T$ and the probe chirp $C_2$, it is possible to access separately the real and the imaginary part of the sum and the difference of the complex REPs**

| | $\bar{T} = n\,T_{vib}$ | $\bar{T} = (n + \frac{1}{4})\,T_{vib}$ |
|---|---|---|
| $C_2 = \frac{m}{2\pi}T_{vib}^2$ | $\Re[R_{g'}^S(\omega) - R_{g'}^{AS}(\omega)]$ | $\Im[R_{g'}^S(\omega) + R_{g'}^{AS}(\omega)]$ |
| $C_2 = \frac{1}{2\pi}\left(m + \frac{1}{4}\right)T_{vib}^2$ | $-\Im[R_{g'}^S(\omega) - R_{g'}^{AS}(\omega)]$ | $\Re[R_{g'}^S(\omega) + R_{g'}^{AS}(\omega)]$ |

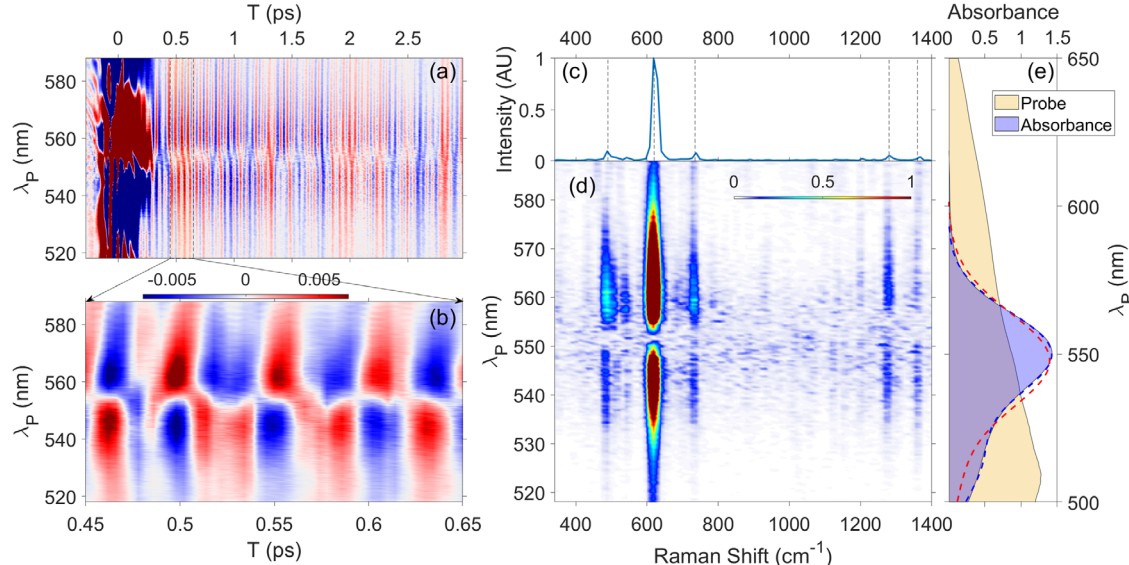

**Fig. 3 | Rhodamine B spectral response.** Time-domain broadband impulsive stimulated Raman scattering spectra of Rhodamine B dissolved in methanol as a function of the probe wavelength $\lambda_P$ recorded with a vanishing probe chirp **a**; a magnification of the oscillating ISRS trace around the 450–650 fs temporal range is reported in **b**. The frequency-domain ISRS spectra obtained upon Fourier transforming over the RP-PP time delay are reported (in arbitrary units) in **d** as a function of $\lambda_P$, while the integrated ISRS spectrum, computed averaging the frequency-domain map over the probe wavelengths from 520 to 585 nm is shown in **c**. A strong

mode at 620 cm$^{-1}$ dominates the signal, with respect to lower intensity peaks at 490, 735, 1280, and 1360 cm$^{-1}$. Small contributions from the solvent and from the glass cuvette may be expected at 1040 and 490 cm$^{-1}$, respectively, and can be suppressed for a vanishing PP chirp. Panel **e** shows the PP spectrum (yellow area) and the sample's absorbance profile (blue area). This latter is compared with the theoretical curves computed via DFT calculations, as discussed in the text, with and without the displacement scaling factor (blue and red dashed lines, respectively).

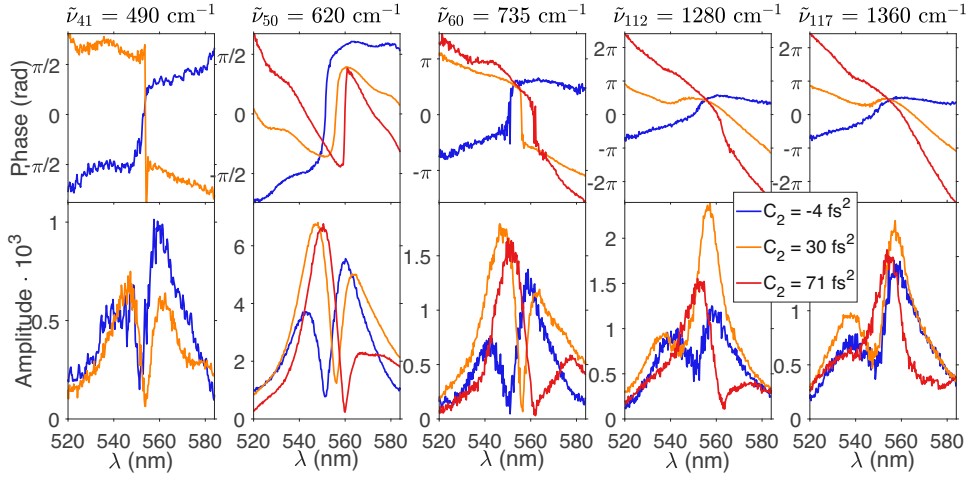

**Fig. 4 | Amplitude and the phase of the experimental ISRS signals.** The amplitude and the phase of the different RhB normal mode components in the ISRS map, recorded for different probe chirp values (reported in the legend), are evaluated across the sample absorption profile as the sum of sinusoidal terms. As expected, for a vanishing PP chirp all the modes exhibit a $\pi$ phase shift around the absorption maximum at 550 nm, where the oscillation amplitudes vanish.

(e, f) for $n = 12$ as a function of the probe wavelength. Specifically, in the chosen reference frame (with $\frac{\partial\alpha}{\partial Q_{g'}} < 0$, cfr. Supplementary Fig. 6), panel (e) shows $-\Re[R_{g'}^S(\omega) - R_{g'}^{AS}(\omega)]$ and $\Im[R_{g'}^S(\omega) + R_{g'}^{AS}(\omega)]$ as continuous green and yellow lines, respectively, while panel (f) $\Im[R_{g'}^S(\omega) - R_{g'}^{AS}(\omega)]$ and $\Re[R_{g'}^S(\omega) + R_{g'}^{AS}(\omega)]$ as blue and purple traces, respectively.

To verify the reconstructed profiles accuracy, the experimental results are compared with the REPs modeled directly from the sample absorption spectrum by using the transform theory[60] (details are reported in the Supplementary Note 5). The computed REPs, which are normalized to the area of the measured traces at $\tilde{T} = 12\,T_{vib}$, are depicted as dashed lines in Fig. 5 and explicitly take into account for the participation of the entire multimode subspace to the Raman response[9], with the adiabatic approximation that is limited to only one dimension, i.e.,

the one of the considered vibration. Building on this approach, when the mixing between ground and excited state normal modes is negligible (as confirmed in Supplementary Note 4 by evaluating the Duschinsky mixing[61]), the unique free parameter required to extract the REPs is the displacement component along the GS normal mode under consideration which we obtained, along with the sign of the molecular polarizability derivative in Eq. (3), by time-dependent (TD) density functional theory (DFT) calculations (details are reported in the Methods section).

As shown in Fig. 5e, f, the normalized profiles and, most importantly, the sign of the measured traces (continuous curves) agree with the modeled signals (dashed lines), validating hence the sign of the reconstructed normal mode ES displacements. Small discrepancies between their lineshapes indicate a possible role of (1) the probe spectral

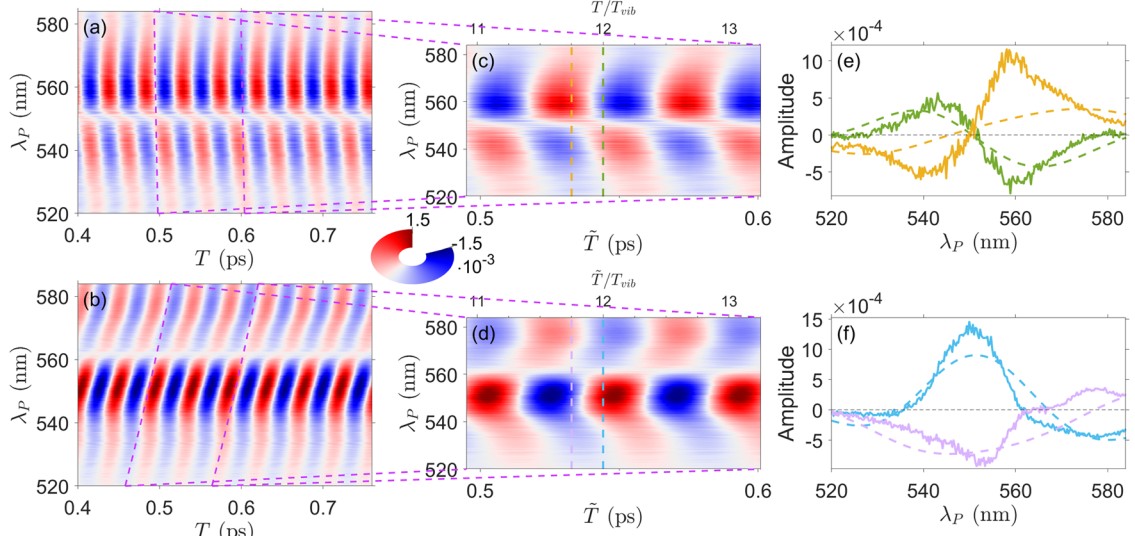

**Fig. 5 | Chirp dependence of the ISRS response and REP reconstruction.** Contributions of the 735 cm$^{-1}$ mode to the ISRS experimental map, measured with an approximately vanishing chirp ($C_2 = -4\,fs^2$) and with a $C_2 = 71\,fs^2 \approx \frac{1}{8\pi} T_{vib}^2$ are reported in **a**, **b**, respectively. The dashed purple lines indicate the slope of the chirp and a magnification of the ISRS maps around 500–600 fs is reported in **c**, **d** as a function of time delay upon dechirp ($\tilde{T}$ bottom axis) and $\tilde{T}/T_{vib}$ (top axis). Slices of such maps at $\tilde{T}/T_{vib} = n$ and $\tilde{T}/T_{vib} = (n - 1/4)$, which allows the complete reconstruction of the real and imaginary parts of the Raman excitation profiles (see

the text and Table 1), are reported (as continuous lines) in **e**, **f** and compared with the REPs modeled by TD-DFT (dashed lines), normalized to the area of the measured traces at $\tilde{T} = 12\,T_{vib}$. Notably, the experimental and the theoretical signals share the same sign, validating hence the sign of the reconstructed normal mode ES displacement, while discrepancies between their spectral profiles suggest a role of the sample absorption, third-order dispersion and probe spectrum, which have not been included in the data analysis reported in this figure.

profile, (2) the sample absorption and (3) the third-order dispersion, which have not been included in the data analysis and should be taken into account aiming at a complete characterization of the complex REPs. Additionally, (4) the molecular ES displacements may deviate with respect the values extracted by time dependent TD-DFT. In fact, (1) since the $S_A$ and $S_B$ signals are generated by interactions with PP spectral components red and blue-shifted with respect to the detected wavelength[52], as reported in Eq. (6) and discussed in the Supplementary Note 1, the reconstructed REPs should be corrected for the corresponding terms $\sqrt{I_P(\omega + \omega_{g'})}$ and $\sqrt{I_P(\omega - \omega_{g'})}$, respectively. Moreover, (2) in presence of a large optical density (up to ~1.5 in the present case, see Fig. 3e), the probe spectral profile strongly varies across the sample, reducing the effective length involved in the signal generation and hence affecting the measured ISRS signal (a complete derivation is discussed in the Supplementary Note 2)[62]. In addition, (3) due to the broad spectral range (>50 nm) under consideration, higher-order dispersion terms should be included to correctly evaluate the arrival time of the different PP spectral components (see Supplementary Note 3).

Finally, in order to evaluate the computed displacements, in Fig. 3e we compared the experimental absorption spectrum with the one obtained from the TD-DFT displacements (blue area and red line, respectively). Notably, the theoretical curve does not accurately reproduce the absorption profile, which in turn can be reproduced by proper scaling the low-frequency mode ($\omega_{g'g} < 1000$ cm$^{-1}$) displacements by a factor 1.2 and the high-frequency ones by 1.4 (blue dashed line in Fig. 3e). In this respect, we note that the relative magnitude of the displacements can be directly extracted from the intensity of the ISRS oscillations, in view of their linear dependence on $|d_j|$. Furthermore, their absolute amplitudes are accessed by taking advantage of the broadband nature of the ISRS detection, which enables a careful mapping of the REPs. The spectral profiles of these latter strongly depend on $d_j$ (details are reported in Supplementary Note 5), giving direct access to the displacement magnitudes.

We further note that the signal-to-noise ratio of the reconstructed REPs can be significantly improved by augmenting the statistical reliability of the extracted signal. To this aim, we reconstruct the REPs

from multiple time delays ($\Delta T \in [400, 700]$ fs with a 4 fs time-step) and PP chirps ($C_2 = -4, 7, 30, 44, 71$ fs$^2$). This can be done by regarding Eq. (6) (Methods section), for each fixed spectral component $\omega$ of the PP, as parametric in $\Delta T$ and $C_2$, with the four unknowns $\Re[R_{g'}^S(\omega)], \Re[R_{g'}^{AS}(\omega)], \Im[R_{g'}^S(\omega)], \Im[R_{g'}^{AS}(\omega)]$. Such over-determined inhomogeneous linear system can be simply solved in a least-squares sense, obtaining the best estimates for the four complex REP terms.

**Absolute excited state geometries**

By using the procedure described above and taking into account for the 1–4 corrections, the complex resonant Raman excitation profiles of the five normal modes under consideration have been extracted and are reported in Fig. 6a. The corresponding excited state displacements with respect to the GS geometry are sketched in 6b and quantitatively reported in Table 2. Since the anti-Stokes REPs are shifted by one vibrational quantum with respect to the Stokes Raman excitation profiles ($R_{g'}^S(\omega) = R_{g'}^{AS}(\omega - \omega_{g'g})$, see Eq. (4)), the former ones have been horizontally shifted for a more direct comparison of the measured traces with the REPs modeled from the absorption spectrum by using the transform method[9,60] and the measured displacements. The experimental traces are in are in good agreement with the theoretical profiles of all the normal modes under investigation, confirming the capability of the presented approach to fully capture the complex REPs and extract the corresponding displacements. Interestingly, our results reveal an increase of the distance between the oxygen and the distal carbon in the RhB central pyran ring, indicating an elongation of this latter in the excited state. Furthermore, the motion of the oxygen nuclei drives an out-of-plane rotation of the carboxylic acid, while a rearrangement of the lateral carbon atoms in the benzene rings leads to an outstretch expansion of these latters. Crucially, as anticipated in the introduction, by fully accessing the real and imaginary part of complex REPs, the reconstructed excitation profiles uniquely determine the sign of excited state displacements along the considered GS eigenvectors $Q_j$. This ultimately unveils the molecular rearrangements in the electronic excited state along the five investigated normal coordinates, which are shown in Fig. 6b.

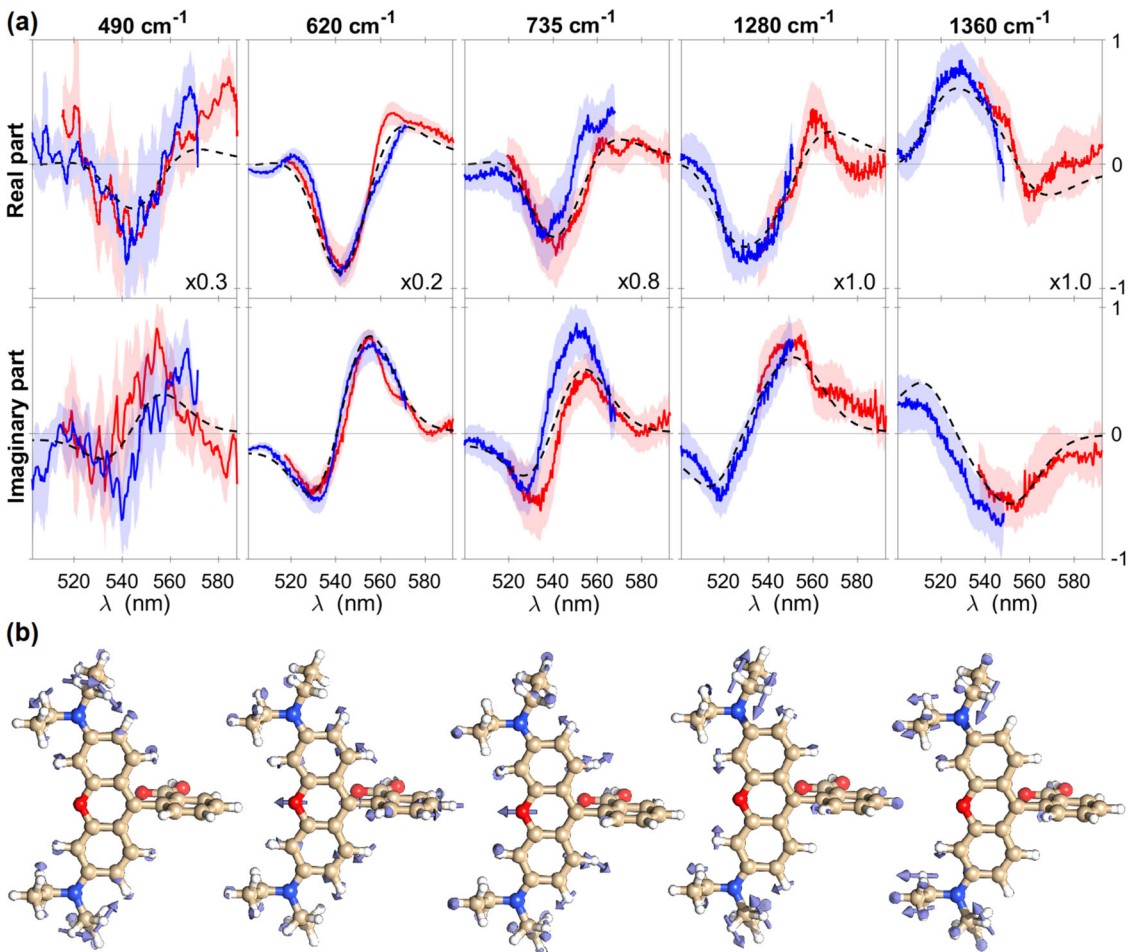

**Fig. 6 | Complex REPs and direction of the ES displacements.** Real and Imaginary part of the Stokes/anti-Stokes (red and blue lines) Raman excitation profiles measured for the different vibrational modes of Rhodamine B dissolved in methanol are reported in **a**. The REPs have been directly extracted from the time-domain traces, from 400 to 700 fs, considering multiple PP chirp values ($C_2 = -4$, 7, 30, 44, 71 fs$^2$). The anti-Stokes Raman excitation profiles have been horizontally shifted by one vibrational quantum for a direct comparison with the Stokes counterparts. Shaded areas indicate the 90% confidence intervals. The experimentally extracted complex REPs are compared with the ones (black dashed lines) modeled from the absorption spectrum by using the transform method[9,60], while the directions of the corresponding ES displacements along the normal coordinates are depicted in **b**.

In summary, building on a diagrammatic treatment of the signal, we have demonstrated that the ISRS response encodes detailed information on both the real and the imaginary parts of molecular Raman excitation profiles. Based on a two-color ISRS scheme with an off-resonant Raman pump and a resonant probe pulse, we have shown how to experimentally measure complex REPs directly in the time-domain, determining the nuclear rearrangements on excited state potential energy surfaces. Notably, since the complex REP of a given normal mode is uniquely determined by two probe chirps and few time delays, the presented approach ensures fast acquisition times and high signal to noise ratios.

The proposed experimental scheme and theoretical model have been benchmarked in order to determine the ES of Rhodamine B dissolved in methanol projected onto different normal modes, identifying an elongation of the central pyran ring in the excited state, driven by an increased the distance between the oxygen and the distal carbon. We anticipate that applying the present protocol to access the ES displacements of photo-active compounds holds the potential for mapping in detail their reaction coordinates, unveiling the first stages of their relaxation dynamics.

## Table 2 | Peak positions and displacements between ground and excited state PESs extracted from the two-color ISRS

Raman excitation profiles, in adimensional form $d = Q\sqrt{\dfrac{m\,\omega_{g'g}}{2\hbar}}$ (column 2) and in atomic units (column 3)

| $\tilde{\nu}_{g'g}$ (cm$^{-1}$) | $d_{EXP}$ | $Q_{EXP}$ (a.u.) | $d_{DFT}$ | $d_{DFT}^{rescaled}$ |
|---|---|---|---|---|
| 490 | −0.12 | −0.049 | −0.12 | −0.15 |
| 620 | −0.24 | −0.058 | −0.20 | −0.24 |
| 735 | −0.08 | −0.020 | −0.14 | −0.17 |
| 1280 | −0.17 | −0.064 | −0.09 | −0.12 |
| 1360 | 0.13 | 0.038 | 0.12 | 0.17 |

In the last columns, the corresponding values obtained via TD-DFT (CAM-B3LYP functional and 6-311++g(2d,2p) basis set) and the ones obtained scaling the computed values to reproduce the sample absorption are reported for comparison.

## Methods

### Experimental setup

The Raman and the probe pulses used for the broadband ISRS experiment are synthesized from the same source, a Ti:sapphire laser, which generates 800 nm transform limited 40 fs pulses, with 1 kHz of repetition rate and an energy of 3.0 mJ. The RP is obtained by an in-house built non-collinear optical parametric amplifier[63], which generates vertically polarized pulses, centered at 700 nm. In order to avoid resonant excitation contributions from the pump, a long-pass filter is used to remove the RP spectral components below 600 nm, while a

pair of chirped mirrors[64] is employed to control the RP compression, ensuring a 18-fs time duration. The vertically polarized broadband PP is synthesized from supercontinuum generation[45] by focusing a small portion of the laser source on a 3 mm sapphire crystal, producing a broadband white light continuum. An additional pair of chirped mirror is exploited to compress the generated PP, whose chirp is fine tuned by introducing thin glass windows on the probe optical path. The RP and the PP are focused in a non-collinear geometry ($\approx 5°$) on the rhodamine sample, which is contained in a 500 μm-thick quartz cuvette. The PP spectrum is monitored on a charge-coupled device upon frequency dispersion by a spectrometer, with a chopper synchronized at 500 Hz that blocks the Raman pump, in order to record consecutive probe pulse spectra, in the presence and in absence of the RP. Further details on the experimental scheme are reported in refs. 38, 52, 65.

### Chirp dependence of the nonlinear ISRS response

The different contributions concurring to the ISRS signal can be computed via the Feynman diagrams introduced in Fig. 2. Specifically, they take into account for two classes of processes, depending on whether the two interactions with the off-resonant Raman pump (RP) involve the ket (Fig. 1a) or the bra-side of the density matrix (Fig. 1b). Among each class, the two interactions with the probe pulse (PP) project the system onto the entire electronic excited state manifold (with a transition amplitude ruled, according to the Franck-Condon principle[24], by the overlap between the ground and excited states vibrational eigenfunctions). Importantly, the off-resonant RP ensures the generation of vibrational coherences $|g'_j\rangle\langle g_j|$ only on the initially populated electronic level for the $A$ diagrams and $|g_j\rangle\langle g'_j|$ for the $B$, modulating the optical properties of the system at the frequencies $\omega^j_{g'g}$ of the stimulated normal modes ($j$). In order to compactly evaluate the measured ISRS response, Eq. (2) can be substituted in Eq. (1), yielding to

$$
\begin{cases}
S(\omega, \Delta T) \propto S_A(\omega, \Delta T) + S_B(\omega, \Delta T) \\
S_A(\omega, \Delta T) = V P^{RP}_{g'} \Re\left[ e^{-i\omega_{g'g}\Delta T} \frac{E_P(\omega - \omega_{g'g})}{E_P(\omega)} R^S_{g'}(\omega)\right] \\
S_B(\omega, \Delta T) = - V P^{RP}_{g'} \Re\left[ e^{i\omega_{g'g}\Delta T} \frac{E_P(\omega + \omega_{g'g})}{E_P(\omega)} R^{AS}_{g'}(\omega)\right]
\end{cases}
\tag{5}
$$

Eq. (5) highlights that, while the vibrational excitations are triggered by the off-resonant Raman pump and are sensitive to the ground state molecular polarizability (via the $\frac{\partial \alpha}{\partial Q_j}$ term in Eq. (3)), the coherences are traced in a pump-probe fashion by the resonant probe pulse, which accesses the vibronic couplings via the Stokes and anti-Stokes Raman excitation profiles. Since in the considered experiments the RP and PP have parallel polarizations, the tensorial nature of the $\alpha$ term can be omitted, being the sample response sensitive only to the isotropic component of the polarizability.

To evaluate the ISRS dependence on the PP chirp[52,55,56,66,67], the probe spectral field can be expressed as $E_P(\omega) = E^0_P(\omega)e^{iC_2(\omega-\omega_P)^2}$, where $E^0_P(\omega)$ indicates the square root of the PP spectrum $I_P(\omega)$.

The ISRS signal can be conveniently recast by introducing the phase term

$$
\phi = \phi(\omega, \Delta T, C_2) = \omega_{g'g}\Delta T + 2 C_2(\omega - \omega_P)\omega_{g'g} = \tilde{T} \cdot \omega_{g'g}
$$

which corresponds to the product between the arrival time of the monitored probe wavelength, i.e., $\tilde{T} = \Delta T + 2 C_2(\omega - \omega_P)$ and the frequency of the considered vibrational mode $\omega_{g'g}$. For a constant probe field (impulsive limit) in the region of interest ($E^0_P(\omega) = E^0_P$), it reads as (the complete derivation is reported in the Supplementary Note 1)

$$
\begin{aligned}
S(\omega, \Delta T) \propto \mathrm{sgn}\left(\frac{\partial \alpha}{\partial Q_{g'}}\right) \cdot \\
\left[\cos\left(C_2\omega^2_{g'g}\right)\left\{\cos(\phi)\Re\left[R^S_{g'}(\omega) - R^{AS}_{g'}(\omega)\right] + \sin(\phi)\Im\left[R^S_{g'}(\omega) + R^{AS}_{g'}(\omega)\right]\right\} + \right. \\
\left. + \sin\left(C_2 \omega^2_{g'g}\right)\left\{\sin(\phi)\Re\left[R^S_{g'}(\omega) + R^{AS}_{g'}(\omega)\right] - \cos(\phi)\Im\left[R^S_{g'}(\omega) - R^{AS}_{g'}(\omega)\right]\right\}\right]
\end{aligned}
\tag{6}
$$

This expression clarifies that tuning the probe chirp $C_2$ to selected values, slices of the dechirped signal (i.e., the ISRS response evaluated as a function of the time delay upon dechirp $\tilde{T}$) can be directly exploited to measure the sum and the difference of the real/imaginary parts of the Stokes/anti-Stokes complex REPs. We stress that, at odd with the spontaneous Raman case, the relative amplitudes of the impulsive Stokes/anti-Stokes Raman responses in Eqs. (2) and (6) do not depend on Boltzmann population factors, since vibrational excitations are stimulated by the off-resonant Raman pump and not by thermal excitation[68].

### Density functional theory calculations

DFT and TD-DFT calculations have been performed with CAM-B3LYP[69] functional and 6-311++g(2d,2p) basis set, taking into account for solvent effect by the polarizable continuum model, by using the Gaussian 16 software package[70]. Upon separate optimization of the ground and the excited state geometries by DFT/TD-DFT, the normal modes $Q_j$ and the associated transition dipole moments have been computed. Separate scans of the molecular geometries along the different GS eigenvectors $Q_j$ are performed to evaluate each molecular polarizability derivative ($\frac{\partial \alpha}{\partial Q_j}$) in the considered reference frame. In parallel, the vertical projection of the ground state PES is monitored exploiting the ground state calculations to generate displaced geometries along the single normal modes under consideration. Further details are provided in the Supplementary Note 8.

## Data availability

All the relevant data are available from the corresponding authors.

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

## Acknowledgements
This project has received funding from the PRIN 2017 Project, Grant No. 201795SBA3-HARVEST (T.S.), from the PRIN 2020 Project, Grant No. 2020HTSXMA-PSIMOVIE (G.B.), from the European Union's Horizon 2020 research and innovation program Graphene Flagship under Grant Agreement No. 881603 (T.S.) and from the NSF Grant CHE-1953045 (S.M.). G.B. and T.S. acknowledge the 'Progetti di Ricerca Medi 2019', the 'Progetti di Ricerca Medi 2020', the 'Progetti di Ricerca Medi 2021' grants by Sapienza Universitá di Roma and the "MESPES: Mapping Excited State Potential Energy Surfaces" and the "VINCIPSI: VIbroNic Couplings In Photoreactive Systems" projects by Italian Super Computing Resource Allocation.

## Author contributions
G.B. and T.S. conceived the study and led the research project. G.B. and E.M. performed the measurements and analysed the data; G.B. and E.M. developed the modeling and carried out the numerical simulations; G.B., E.M., G.F., S.M and T.S. discussed the results and wrote the manuscript.

## Competing interests
The authors declare no competing interests.
