## [Peer Review File · Nature Communications]

REVIEWER COMMENTS

Reviewer #1 (Remarks to the Author):

The paper describes a two-dimensional Raman spectroscopic experiment on rhodamine

B and a theory is applied for its interpretation. I think that the data is interesting and worthy for publication, but worry about the following points.

- the introduction is nice, but little bit unfocused:

- talking about RR cross section, the authors probably want to say that it is proportional to the square of both electronic transition moment and the vibrational overlap

- how do they define the “sign” of the GS-> ES geometry change? For me the sign of a normal mode is quite arbitrary and the whole discussion makes no sense, perhaps direction would be a better term.

- the equations presented seem to ignore the tensor (alpha)/vector(dipole, el. field) character of the variables, why? Overall, the equations are difficult to follow. Perhaps it might help to separate more the theory (to SI) and in the text clearly define what and how was measured. In the theory then point out the most important steps...

- Figure 1a, I am missing the difference between Stokes and anti-Stokes, are the arrows correct?

Reviewer #2 (Remarks to the Author):

Referee report on the MS NCOMMS-22-25315

“Absolute excited state molecular geometries revealed by resonance Raman signals”

by Giovanni Batignani, Emanuele Mai, Giuseppe Fumero, Shaul Mukamel, and Tullio Scopigno

submitted to Nature Communications

Electronic excitations of molecules are crucial in such diverse fields and applications like photosynthesis, the vision process, OLEDs, and organic photovoltaics, to name a few. Obviously, these excitations alter the distribution of electrons within the molecule. The nuclei respond to that and (slightly) alter their equilibrium positions, i.e. the molecular geometries change. As with molecules in their electronic ground state, these geometries impact their physical and chemical

properties and, thus, knowledge about them is of utmost importance. In the ground state, molecular structures can be deduced from diffraction and NMR experiments. For excited states their short lifetime (nanosecond or shorter) poses a challenge. Even though time resolved diffraction methods show impressive progress, for molecular structures of excited states one usually has to rely on spectroscopic signature.

Changes of collective nuclear coordinates (normal modes) upon electronic excitation define the band shape of the respective electronic transition (Franck Condon (FC) factors). FC factors can also be determined by resonance Raman (RR) spectroscopy. In an RR spectrum, the signal strength at a wavenumber associated with a particular normal mode is proportional to a certain FC factor squared. From this factor the geometric change along the normal can be computed. Yet, since the square of this factor is measured, the sign information is lost. Therefore, based on RR experiments one cannot decide whether a chemical bond is shortened or lengthened upon excitation or whether an angle gets smaller or larger. Chemical intuition can help but an experimental answer is of course preferred.

In the present contribution, the authors design, realize, and apply an experimental technique which provides this answer. In the technique, a non-resonant femtosecond laser pulse triggers vibrational coherence of a dye molecule (Rhodamine B). These coherences are then traced in pump-probe fashion by a probe pulse in resonance with the $S_0 \rightarrow S_1$ transition. Analysis of the oscillatory features in the recorded time traces finally yields magnitude and sign of the displacement of along normal coordinates. These displacements compare very favorably with predictions from quantum chemistry. I find this a very impressive achievement which combines an elaborate theoretical analysis (see, however, below), a challenging and beautifully conducted experiment and a result which was cross-checked by a completely different approach (quantum chemistry). Due to that and the relevance of excited state structures (see above), the paper should appear in a high impact journal like Nature Communications. Prior to that the paper should be revised according to the below issues.

Major issues

(1) I am not a complete stranger to femtosecond techniques and Raman spectroscopy. Yet, I really struggled to follow the line of argument which explains why the presented technique yields the sign of the displacement. I am still not sure whether I got right. As a journal like Nature Communications addresses a very broad audience, the authors should really make a huge effort to better explain their approach. I could, for instance, imagine that a graphic which illustrates which of the relevant physical quantities change sign with the displacement and why could help.

(2) With their femtosecond experiments the authors covered Raman resonances up to $\sim 1400 \text{ cm}^{-1}$. This is of course related to the pulse durations and should be mentioned.

(3) The authors studied the influence of the probe pulse chirp theoretically and experimentally. What I did not quite understand: Are experiments with different chirps necessary to arrive at the sign of the displacements?

(4) From conventional RR spectroscopy, I recall that absolute cross sections are necessary to arrive at the (unsigned) displacements. How is the situation here?

Minor issues

(a) In the Abstract and earlier on in the Introduction stress the connections between the displacements upon excitation and biological activity. This might give the completely wrong impression that such displacements are not observed or relevant for other molecules.

(b) I suggest avoiding the terms Stokes and anti-Stokes throughout the paper and particular in Figure 1. In Figure 1a, the vertical green arrow starts from a vibrationally excited state, so in conventional Raman this would be termed anti-Stokes. As the signal is measured on the low frequency side the term Stokes is used. To avoid this confusion, the terms should not be used.

(c) In the caption to Figure 1, it should be mentioned whether the vertical arrows refer to electric fields or photons.

(d) On page 8, the authors introduce the frequency difference ω_{ji} . In the eq. (2) and later on the symbol $\omega_{g'g}$ is used. Why?

(e) In the SI, the authors describe a Duschinsky analysis of the mode mixing. It should be mentioned in the main text why this analysis was performed and what the essential result is.

(f) In the Introduction I miss reference(s) to the Anne Myers' work on RR spectroscopy.

Reviewer #1 (Remarks to the Author):

The paper describes a two-dimensional Raman spectroscopic experiment on rhodamine B and a theory is applied for its interpretation. I think that the data is interesting and worthy for publication, but worry about the following points.

We thank the Referee for appreciating the interest and worth for publication of our work. We have done our best in the revised version to address all the issues raised, which we found very constructive.

- the introduction is nice, but little bit unfocused:

We made our best to refocus the introduction (please see the attached document, where all the differences with respect to the previous version are highlighted). The main changes are:

- We have clarified that the nuclear rearrangement undergoing along the displaced normal modes is crucial for determining the physical/chemical properties on the ES.
- We have clarified that our results unambiguously identify the direction of the ES rearrangement.
- The multidimensional displacement vector has been defined: “the vector identifying the new equilibrium position in the ES projected onto the eigenvectors (\mathbf{Q}_j) of the GS with respect to its minimum. Large absolute values of specific displacement components ($d_j = \mathbf{D} \cdot \mathbf{Q}_j$, commonly referred to as *displacements*) identify the vibrational eigenmodes which are more involved in the reaction and eventually their couplings to the specific electronically excited state”.
- The discussion on resonant Raman spectroscopy has been refocused and it is now more concise.

- talking about RR cross section, the authors probably want to say that it is proportional to the square of both electronic transition moment and the vibrational overlap.

We agree that mentioning the cross-section dependence on electronic transition moment and the vibrational overlap is in order. This point has been fixed, clarifying that “Tuning the excitation wavelength to match the energy of an allowed electronic transition to a given ES enables resonant Raman (RR) spectroscopy and results in a cross section proportional to the square modulus of the product between the electronic transition moments and the vibrational GS-ES overlaps”.

- how do they define the “sign” of the GS-> ES geometry change? For me the sign of a normal mode is quite arbitrary and the whole discussion makes no sense, perhaps direction would be a better term.

We thank the Referee for the comment. In the revised version of the manuscript, we stressed more clearly that our approach provides the chance to unambiguously determine the magnitude and the direction (intended as orientation and sense) of the nuclear rearrangements between ground and excited states along the considered normal coordinates. This allows us to identify the absolute position of the ES PES minima.

For what concerns assessing whether the displacements are positive or negative with respect to ground state eigenvectors orientations, we realized that a clarification is in order. As the Referee correctly pointed out, the assignment of the sign (sense) for a given eigenvector, obtained by diagonalizing the Hessian matrix, is arbitrary in the sense that the arrows identifying the direction of the atomic vibrations may be reversed. However, the invariance of the ISRS response on the choice of such a reference frame is restored by the dependence of the measured signals on both the FC factors and the polarizability derivative along the normal coordinate $\partial\alpha/\partial Q$, which is not vanishing

for a Raman active vibration (see Eqs. 3-5): the sign of both these quantities are reversed by changing the sense of the GS eigenvector under consideration.

Having defined the direction of the GS eigenvectors (for example, all the directions may be chosen in a way that $\frac{\partial\alpha}{\partial Q} > 0$), the sign of the ES displacements relative to such a reference frame can be unambiguously defined. Finally, building on the measured REPs, the nuclear rearrangements on excited state PESs are then identified, ultimately revealing the ES geometry.

These points have been clarified in the revised version of the manuscript, further stressing that our results ultimately determine absolute ES molecular geometries. Following a suggestion by Referee # 2, this is further discussed in a new figure of the manuscript (Fig. 1, also reported below), where we illustrate the application of the presented approach to the exemplificative case of a diatomic molecule (please see our reply to the first comment of Referee 2).

Figure R1: Concept of the proposed ISRS scheme for a diatomic molecule. A pair of ultrashort pulses is exploited to measure complex REPs and access the ES displacement, determining whether the atomic distance is increased or decreased upon the photoexcitation (panels a-b). Since the ISRS response is proportional to the product between the GS polarizability derivative and the FC overlaps, the measured signals $S(\lambda)$ uniquely determines the ES geometry and the sense of the geometrical rearrangement (panels c-d), being antisymmetric with respect to the nuclear distance modification. Notably, while the sense of the GS eigenvector is arbitrary, as both outgoing (\mathbf{Q}) or in-going arrows ($-\mathbf{Q}$) can be selected (panel e), reversing such a sense changes both the sign of the polarizability derivative as well as the sign of the FC overlaps (panels f-g), making the experimental signals invariant on the selection of the reference frame.

- the equations presented seem to ignore the tensor (alpha)/vector(dipole, el. field) character of the variables, why? Overall, the equations are difficult to follow. Perhaps it might help to separate more the theory (to SI) and in the text clearly define what and how was measured. In the theory then point out the most important steps.

Following the Referee suggestion, we moved some of the equations (in particular, the ones focusing on the derivation of the ISRS chirp dependence) to the Methods section, discussing in the main the theoretical background and the link with the experiments. We also apologize for the lack of clarity

on the tensorial nature of the polarizability term. As we have now clarified in the revised version of the paper, equations ignore its tensorial nature due to (1) the isotropic nature of the investigated sample (Rhodamine dissolved in methanol) and (2) the parallel polarization of the Raman and Probe fields. Under such conditions, the ISRS response depends only on the isotropic component of the polarizability tensor, while in more general cases the tensorial nature may be included straightforwardly in Eqs. 3-5.

- *Figure 1a, I am missing the difference between Stokes and anti-Stokes, are the arrows correct?*

We confirm that the labelling is correct, but, after the Referee comment, we realized that the color of the green arrow indicating the third light-matter interaction was too close to the color of the Probe field and this could have been misleading for the readers. We revised the colors and the appearance of the Probe envelope to solve this issue.

For what concerns the difference between the two pathways: in the Stokes processes two (off resonant) interactions between the Raman pump and ket side of the molecular density matrix stimulate the vibrational coherence. Then, an interaction with the broadband resonant probe (still on the ket side) promotes the system to the electronic excited state manifold, and is followed by a free induction decay which enables to record the Raman response. Conversely in the anti-Stokes pathways, while the interaction with the probe remains in the ket side of the density matrix, the interactions with the Raman pump occur on the bra, leading to a response sensitive to the anti-Stokes REP (see Eq. 4).

Reviewer #2 (Remarks to the Author):

Electronic excitations of molecules are crucial in such diverse fields and applications like photosynthesis, the vision process, OLEDs, and organic photovoltaics, to name a few. Obviously, these excitations alter the distribution of electrons within the molecule. The nuclei respond to that and (slightly) alter their equilibrium positions, i.e. the molecular geometries change. As with molecules in their electronic ground state, these geometries impact their physical and chemical properties and, thus, knowledge about them is of utmost importance. In the ground state, molecular structures can be deduced from diffraction and NMR experiments. For excited states their short lifetime (nanosecond or shorter) poses a challenge. Even though time resolved diffraction methods show impressive progress, for molecular structures of excited states one usually has to rely on spectroscopic signature. Changes of collective nuclear coordinates (normal modes) upon electronic excitation define the band shape of the respective electronic transition (Franck Condon (FC) factors). FC factors can also be determined by resonance Raman (RR) spectroscopy. In an RR spectrum, the signal strength at a wavenumber associated with a particular normal mode is proportional to a certain FC factor squared. From this factor the geometric change along the normal can be computed. Yet, since the square of this factor is measured, the sign information is lost. Therefore, based on RR experiments one cannot decide whether a chemical bond is shortened or lengthened upon excitation or whether an angle gets smaller or larger. Chemical intuition can help but an experimental answer is of course preferred. In the present contribution, the authors design, realize, and apply an experimental technique which provides this answer. In the technique, a non-resonant femtosecond laser pulse triggers vibrational coherence of a dye molecule (Rhodamine B). These coherences are then traced in pump-probe fashion by a probe pulse in resonance with the $S_0 \rightarrow S_1$ transition. Analysis of the oscillatory features in the recorded time traces finally yields magnitude and sign of the displacement of along normal coordinates. These displacements compare very favorably with predictions from quantum chemistry.

I find this a very impressive achievement which combines an elaborate theoretical analysis (see, however, below), a challenging and beautifully conducted experiment and a result which was cross-checked by a completely different approach (quantum chemistry). Due to that and the relevance of excited state structures (see above), the paper should appear in a high impact journal like Nature Communications. Prior to that the paper should be revised according to the below issues.

We thank the Referee for appreciating our work, stressing the potential impact of the presented results in the physical-chemistry field and recommending publication in Nature Communications. We really appreciated Her/His comments, which provided us the chance to further improve our manuscript.

Major

issues

(1) I am not a complete stranger to femtosecond techniques and Raman spectroscopy. Yet, I really struggled to follow the line of argument which explains why the presented technique yields the sign of the displacement. I am still not sure whether I got right. As a journal like Nature Communications addresses a very broad audience, the authors should really make a huge effort to better explain their approach. I could, for instance, imagine that a graphic which illustrates which of the relevant physical quantities change sign with the displacement and why could help.

We thank the Referee for the suggestion. We realized a new Figure to illustrate the concept of our work and how the sign of the ES displacements can be determined from the measured quantities (attached Figure R1). Their dependence on the relevant physical quantities is also discussed.

The concept is illustrated for the illustrative case of a diatomic molecule, where the unique degree of freedom is the distance between the nuclei, which can be either increased or decreased on the ES (top left panel). As shown in the central panels (c and d), the sign of the ISRS signals (and of the

reconstructed complex REPs) shows an odd symmetry depending on the sign of the atomic distance modification with respect to the GS (the response is evaluated for the same displacement magnitude of the two ES geometries).

Figure R1: Concept of the proposed ISRS scheme for a diatomic molecule. A pair of ultrashort pulses is exploited to measure complex REPs and access the ES displacement, determining whether the atomic distance is increased or decreased upon the photoexcitation (panels a-b). Since the ISRS response is proportional to the product between the GS polarizability derivative and the FC overlaps, the measured signals $S(\lambda)$ uniquely determines the ES geometry and the sense of the geometrical rearrangement (panels c-d), being antisymmetric with respect to the nuclear distance modification. Notably, while the sense of the GS eigenvector is arbitrary, as both outgoing (Q) or in-going arrows ($-Q$) can be selected (panel e), reversing such a sense changes both the sign of the polarizability derivative as well as the sign of the FC overlaps (panels f-g), making the experimental signals invariant on the selection of the reference frame.

For what concerns assessing whether the displacements are positive or negative with respect to ground state eigenvectors orientations, we realized that a clarification is in order. As the Referee correctly pointed out, the assignment of the sign (sense) for a given eigenvector, obtained by diagonalizing the Hessian matrix, is arbitrary in the sense that the arrows identifying the direction of the atomic vibrations may be reversed. However, the invariance of the ISRS response on the choice of such a reference frame is restored by the dependence of the measured signals on both the FC factors and the polarizability derivative along the normal coordinate $\partial\alpha/\partial Q$, which is not vanishing for a Raman active vibration (see Eqs. 3-5): the sign of both these quantities are reversed by changing the sense of the GS eigenvector under consideration.

(2) With their femtosecond experiments the authors covered Raman resonances up to $\sim 1400 \text{ cm}^{-1}$. This is of course related to the pulse durations and should be mentioned.

Correct. Following the Referee suggestion this point has now been clarified: “a pair of chirped mirrors [65] is employed to control the RP compression, ensuring a 18-fs time duration. For such a duration, vibrational modes up to 1400 cm^{-1} can be observed by the ISRS technique [30].”

(3) The authors studied the influence of the probe pulse chirp theoretically and experimentally. What I did not quite understand: Are experiments with different chirps necessary to arrive at the sign of the displacements?

We thank the Referee for this comment. As we have now clarified in the revised version of the manuscript, the proposed scheme can be exploited to identify the sign of the ES displacements along the investigated normal modes by performing a single measurement (ideally with a vanishing chirp). This is due to the odd symmetry of the spectral dependence of the ISRS signals as a function of the displacements (reported in panel e of Figure 2). However, in order to determine the Raman excitation profiles, at least four different measurements (with 2 different probe chirps and 2 different time delays between pump and probe) must be performed (see table 1).

(4) From conventional RR spectroscopy, I recall that absolute cross sections are necessary to arrive at the (unsigned) displacements. How is the situation here?

This is an interesting comment. The Referee is indeed correct, the normalization of different spectra (acquired changing the wavelength of the continuous wave pump) can be an issue for conventional RR. Here, the broadband nature of the probe pulse provides the chance to record the Raman excitation profiles over all the sample absorption in a single measurement. Hence normalization is not an issue. Specifically, in view of the heterodyne detection, the ISRS signal directly accesses the Raman response as the differential intensity of probe intensity (conventionally referred as Raman Gain, i.e. the ratio of probe intensity in presence and in absence of the Raman pump): the experimental response depends uniquely on the effective pathlength (sample thickness and concentration), product of GS/ES dipole matrix elements (ES displacement), molecular polarizability and pump fluence.

For large displacements ($|d| > 0.2$), this provides the chance to exploit the spectral dependence of the (signed) REP amplitudes to directly obtain the (signed) displacements: this is illustrated in Figure S4 of the Supporting Information, where the normalized REP lineshapes are reported as a function of d . In this case absolute cross sections are hence not necessary. We also note that for small amplitudes of the ES displacement ($|d| \leq 0.2$), which is actually the case of vibrational modes barely involved in ES rearrangements, the normalized REPs become less sensitive to the amplitudes of d ; under such regime, the relative amplitudes of the displacements along the different normal modes can be obtained from the relative intensities of the ISRS signals; then the displacement magnitudes can be retrieved by simultaneously fitting the absorption profile.

Minor

issues

(a) In the Abstract and earlier on in the Introduction stress the connections between the displacements upon excitation and biological activity. This might give the completely wrong impression that such displacements are not observed or relevant for other molecules.

We agree with the Referee, ES displacements can be relevant also for non-biologically photo-active molecules. In the revised version we have clarified that ES displacements more in general determine physical/chemical properties and that they are crucial for studying both biologically active systems and photochemical events.

(b) I suggest avoiding the terms Stokes and anti-Stokes throughout the paper and particular in Figure 1. In Figure 1a, the vertical green arrow starts from a vibrationally excited state, so in conventional Raman this would be termed anti-Stokes. As the signal is measured on the low frequency side the term Stokes is used. To avoid this confusion, the terms should not be used.

We agree and we recognize that the classification of stimulated Raman processes as either Stokes or anti-Stokes can be ambiguous; in fact, according to the most common nomenclature, the distinction between Stokes and anti-Stokes processes is based on the temporally last Raman interaction. One speaks of a Stokes process when, due to the last interaction, the molecule passes from a state of lower to higher energy and the emission is red-shifted; conversely in anti-Stokes the last transition is from a state of higher energy to one of lower energy and the emission is blue-shifted (the term coherent anti-Stokes Raman spectroscopy comes from this convention). One can alternatively distinguish Stokes and anti-Stokes processes by looking at the overall energy transfer between the matter and the fields, during the entire process with all interactions, not just the last. Stokes processes start from the ground state, while anti-Stokes processes start from vibrationally excited states. Another alternative is to distinguish Stokes and anti-Stokes processes by looking at the dependence of the Raman response to the probed wavelength: in the impulsive Raman case, due to the excitation of the vibrational coherence induced by a pump field different with respect to the probe, Feynman pathways are sensitive to either Stokes or anti-Stokes resonant Raman excitation profiles of spontaneous Raman spectroscopy (see Eq. 2,4).

To avoid these ambiguities, following the Referee suggestion, we decided not to indicate the diagrams of Figure 1 as Stokes or anti-Stokes, neither to refer to Stokes/anti-Stokes polarizabilities. We then clarified that: (i) the cross section of the Feynman diagrams a/b is sensitive to the Stokes/anti-Stokes Raman excitation profile, respectively, and (ii) at odd with the spontaneous Raman case, the amplitude of the impulsive Raman response does not depend on the Boltzmann population factor, since vibrational coherences are generated by the off resonant Raman pump and not by thermal excitation.

(c) In the caption to Figure 1, it should be mentioned whether the vertical arrows refer to electric fields or photons.

We agree, now we have clarified in the caption of Figure 1 that “Interactions with the bra/ket side of the density matrix and the (classical) electromagnetic fields are presented by dashed/continuous vertical arrows, respectively”.

(d) On page 8, the authors introduce the frequency difference ω_{ji} . In the eq. (2) and later on the symbol $\omega_{g'g}$ is used. Why?

We apologize for the lack of clarity; the ω_{ji} notation has been used to generically indicate the frequency difference between two arbitrary (either electronic or vibrational) levels $j-i$. This notation is then used on page 8 for discussing the pump-induced vibrational coherences (Eqs. 2-3) and later on for evaluating the electronic ground vs excited state vibronic couplings via the interaction with the probe pulse. In the original version of the manuscript, we actually referred to ω_{ji} as the frequency difference between vibrational levels; this typo has been corrected.

(e) In the SI, the authors describe a Duschinsky analysis of the mode mixing. It should be mentioned in the main text why this analysis was performed and what the essential result is.

We apologize for the lack of clarity. In presence of Duschinsky rotation, the excited state eigenvectors are a linear combination of multiple ground state ones. Under such condition, a theoretical evaluation of the REPs requires as input parameters both the normal mode displacements and their

mixing angles. Ruling out mode mixings between the ground and the excited states ensures that the unique free parameters required to extract the REPs are the displacements along the GS normal mode under consideration. This point has been clarified in the manuscript.

(f) In the Introduction I miss reference(s) to the Anne Myers' work on RR spectroscopy.

We thank the Referee for bringing our attention to the interesting works by Anne Myers that provide further illustrative examples on the importance of Raman Excitation Profiles and are useful to stress the impact of our work with respect to the determination of the sign of ES displacements. Appropriate references ("Excited state geometry changes from preresonance Raman intensities: Isoprene and hexatriene", "Resonance Raman Intensities and Charge-Transfer Reorganization Energies", "Femtosecond molecular dynamics probed through resonance Raman intensities", "Resonance Raman and Resonance Hyper-Raman Intensities: Structure and Dynamics of Molecular Excited States in Solution", and "Resonance Raman and photoluminescence excitation profiles and excited-state dynamics in CdSe nanocrystals") have now been included in the bibliography.

REVIEWERS' COMMENTS

Reviewer #1 (Remarks to the Author):

I think that the revised manuscript is suitable for the publication now. It introduces a new technique that may be of interest for a broad audience.

Reviewer #2 (Remarks to the Author):

Referee report on the revised MS NCOMMS-22-25315 A

“Absolute excited state molecular geometries revealed by resonance Raman signals”

by Giovanni Batignani, Emanuele Mai, Giuseppe Fumero, Shaul Mukamel, and Tullio Scopigno
submitted to Nature Communications

The authors have responded to the scientific and technical issues addressed by the reviewers in a satisfactory way. The major concern of reviewer #1 and myself (reviewer #2) was the readability of the manuscript. With their revision, the authors have improved on. In this respect, I really like the new Figure 1. However, I find the paper – in particular from page 8 on-wards - still difficult to follow. I understand and appreciate that for scientific rigor the authors have to present the mathematical formalism and that this can be in conflict with the readability. Maybe by discussing the paper with a colleague not involved in the study, the authors can further improve on that. I would really like this very nice study to be as readable as possible.

Suggestions in that direction from my side:

1) As I have written above, I really like the new Figure 1. However, in the text (page 8, first paragraph) they mostly use it to discuss the sign convention – in response to reviewer 1. They only touch on my concern, namely how an increase or decrease of a bond-length affects the Raman signature here. They should expand a bit on that.

2) Concerning anti-Stokes / Stokes: Figure 2 and throughout the text. As the authors have admitted in their response letter, a definition is a bit involved in the present context. So why not dropping it completely and simply refer to the terms A and B - as it is partially done already? They might also add a remark about why using the terms anti-Stokes / Stokes here is problematic.

Minor comments:

a) On page 1, I suggest changing “The nuclei respond to photo-excitation altering their equilibrium positions and the molecule undergoes a geometrical rearrangement.” to “The nuclei respond to photo-excitation altering their equilibrium positions, i.e. the molecule undergoes a geometrical rearrangement.”

b) On page 4, the authors state “...the vibrational eigenmodes which are more involved in the reaction and eventually”. I suggest replacing “reaction” by “excitation”. A reaction implies chemistry and the FC active modes are not necessarily the ones involved in a photo-reaction.

c) The abbreviation “REP” (Raman excitation profile?) used throughout the text is not defined.

d) The influence of the duration of the pump pulse on the spectral coverage is quite important (see my first report) and should not be “hidden” in the Methods section.

e) In the caption to Figure 6 it should be mentioned that the black dashed lines refer to results from TD-DFT computations.

REVIEWERS' COMMENTS

Reviewer #1 (Remarks to the Author):

I think that the revised manuscript is suitable for the publication now. It introduces a new technique that may be of interest for a broad audience.

We sincerely thank the Referee for appreciating our work and recommending for publication.

Reviewer #2 (Remarks to the Author):

Referee report on the revised MS NCOMMS-22-25315 A

“Absolute excited state molecular geometries revealed by resonance Raman signals”

by Giovanni Batignani, Emanuele Mai, Giuseppe Fumero, Shaul Mukamel, and Tullio Scopigno submitted to Nature Communications

The authors have responded to the scientific and technical issues addressed by the reviewers in a satisfactory way. The major concern of reviewer #1 and myself (reviewer #2) was the readability of the manuscript. With their revision, the authors have improved on. In this respect, I really like the new Figure 1. However, I find the paper – in particular from page 8 on-wards - still difficult to follow. I understand and appreciate that for scientific rigor the authors have to present the mathematical formalism and that this can be in conflict with the readability. Maybe by discussing the paper with a colleague not involved in the study, the authors can further improve on that. I would really like this very nice study to be as readable as possible. Suggestions in that direction from my side:

We thank again the Referee for appreciating our work, recommending for publication. We have done our best to make the paper as readable as possible, by following the comments/suggestions below, by separating the Results section in different subsection, and by further simplifying all the technical details (which are discussed in detail in the Methods and in the SI).

1) As I have written above, I really like the new Figure 1. However, in the text (page 8, first paragraph) they mostly use it to discuss the sign convention – in response to reviewer 1. They only touch on my concern, namely how an increase or decrease of a bond-length affects the Raman signature here. They should expand a bit on that.

We thank the Referee for the suggestion. A more detailed description of the relation between the experimental signal and the bond-length modification in the ES has now been included. Indeed, it is now detailed with an example that the measured complex REP yields the same physical observable (i.e. the bond length modification), independently of the chosen definition of the eigenmode direction.

The following specifications have been added in the text:

- “Importantly, ES molecular geometries corresponding to an increase or a decrease of the nuclear bond-length have FC overlaps with the same magnitude and opposite sign. Hence, the ISRS signal, recorded with the two-colour non degenerate scheme presented in this work (panels b), is antisymmetric with respect to the GS-to-ES atomic distance modification (panels c-d) and directly reveals the ES displacement.”

Further details are also provided in the caption of Figure 1:

- “For example, if the complex REP reconstructed from the ISRS signal is the one reported in panel (c), the consequences of the two possible eigenmode choices (e) are depicted in the corresponding panel (f): the eigenmode direction \mathbf{Q} is associated to a positive $\frac{\partial\alpha}{\partial Q}$ and a positive displacement d , while the direction $\mathbf{Q}' = -\mathbf{Q}$ corresponds to a negative polarizability derivative as well as to a negative d . Crucially, the two formal descriptions yield to the same physical observable, i.e. an increase of the bond length in the excited state (highlighted in red). On the other hand, the measured REP in panel (c) excludes an ES bond-length shortening (blue), as the corresponding descriptions shown in panel (g) are both associated to the signal in panel (d), which has a reversed sign with respect to the measured one.”

2) Concerning anti-Stokes / Stokes: Figure 2 and throughout the text. As the authors have admitted in their response letter, a definition is a bit involved in the present context. So why not dropping it completely and simply refer to the terms A and B - as it is partially done already? They might also add a remark about why using the terms anti-Stokes / Stokes here is problematic.

Following the Referee suggestion, we clarified that the A/B pathways reported in the diagrams of Figure 2 cannot be identified as spontaneous Raman Stokes/anti-Stokes processes. For what concerns the Raman excitation profiles ($R_{g,r}^S(\omega)$ $R_{g,r}^{AS}(\omega)$) defined in Eq. 4 and extracted from the presented ISRS scheme, we preserved the S/AS labelling, since the square moduli of the extracted quantities are precisely equivalent to the Stokes/anti-Stokes excitation profiles accessed by spontaneous Raman spectroscopy. Also, in view of the broad readership of Nature Communications, we believe that preserving the S/AS notation is important for linking some of the information, extracted with the coherent Raman approach presented here, to experimental observables conventionally investigated by the spontaneous Raman readership. These points have been clarified in the revised manuscript after Eq. 4: “Importantly, while the A/B diagrams reported in Figure 2(a-b) do not correspond to the spontaneous Raman Stokes/anti-Stokes processes, the square moduli of the REP reported in Eq. 4 are the ultimate information that can be accessed by spontaneous Raman spectroscopy [11].”

A more detailed discussion on the use of Stokes vs anti-Stokes terminology has been included in the SI: “It is worth to stress that we avoided referring to stimulated Raman pathways A/B as either Stokes or anti-Stokes since such a classification can be ambiguous. In fact, according to the most common nomenclature, the distinction between Stokes and anti-Stokes processes is based on the temporally last Raman interaction. One speaks of a Stokes process when, due to the last interaction, the molecule passes from a state of lower to higher energy and the emission is red-shifted; conversely in anti-Stokes the last transition is from a state of higher energy to one of lower energy and the emission is blue-shifted (the term coherent anti-Stokes Raman spectroscopy comes from this convention). One can alternatively distinguish Stokes and anti-Stokes processes by looking at the overall energy transfer between the matter and the fields, during the entire process with all interactions, not just the last. Stokes processes start from the ground state, while anti-Stokes processes start from vibrationally excited states. Another alternative is to distinguish Stokes and anti-Stokes processes by looking at the dependence of the Raman response to the probed wavelength: in the impulsive Raman case, due to the excitation of the vibrational coherence induced by a pump field different with respect to the probe, Feynman pathways are sensitive to either Stokes or anti-Stokes resonant Raman excitation profiles of spontaneous Raman spectroscopy (see Eq. 2,4 of the main text).”

Minor comments:

a) On page 1, I suggest changing “The nuclei respond to photo-excitation altering their equilibrium positions and the molecule undergoes a geometrical rearrangement.” to “The nuclei respond to photo-excitation altering their equilibrium positions, i.e. the molecule undergoes a geometrical rearrangement.”

We thank the Referee for the suggestion. The text has been modified accordingly.

b) On page 4, the authors state “...the vibrational eigenmodes which are more involved in the reaction and eventually”. I suggest replacing “reaction” by “excitation”. A reaction implies chemistry and the FC active modes are not necessarily the ones involved in a photo-reaction.

We thank the Referee for the suggestion. The text has been modified accordingly.

c) The abbreviation “REP” (Raman excitation profile?) used throughout the text is not defined.

We thank the Referee for pointing out this typo. The acronym for Raman excitation profile (REP) has now been defined where it first appears in the text, as well as in the caption of Fig.1.

d) The influence of the duration of the pump pulse on the spectral coverage is quite important (see my first report) and should not be “hidden” in the Methods section.

We moved the comment from the Methods section to the Results section, along with the description of the different Raman modes detected via the Fourier transform shown in Fig 1.(d). The text now reads:

“By fast Fourier transforming over the RP-PP time delay, the frequency-domain spectrum (reported in Figure 2 c-d) is extracted, identifying a strong mode at 620 cm^{-1} and other weaker Raman bands at 490, 735, 1280 and 1360 cm^{-1} . Indeed, the $\sim 18\text{ fs}$ temporal duration of the pump pulse guarantees efficient stimulation of coherences up to 1400 cm^{-1} .”

e) In the caption to Figure 6 it should be mentioned that the black dashed lines refer to results from TD-DFT computations.

We thank the Referee for the note, which has been implemented.